# Unveiling the radiative local density of optical states of a plasmonic nanocavity by STM

Alberto Martín-Jiménez[1], Antonio I. Fernández-Domínguez [2], Koen Lauwaet[1], Daniel Granados[1], Rodolfo Miranda[1,3], Francisco J. García-Vidal[2,4 ✉] & Roberto Otero [1,3 ✉]

Atomically-sharp tips in close proximity of metal surfaces create plasmonic nanocavities supporting both radiative (bright) and non-radiative (dark) localized surface plasmon modes. Disentangling their respective contributions to the total density of optical states remains a challenge. Electroluminescence due to tunnelling through the tip-substrate gap could allow the identification of the radiative component, but this information is inherently convoluted with that of the electronic structure of the system. In this work, we present a fully experimental procedure to eliminate the electronic-structure factors from the scanning tunnelling microscope luminescence spectra by confronting them with spectroscopic information extracted from elastic current measurements. Comparison against electromagnetic calculations demonstrates that this procedure allows the characterization of the meV shifts experienced by the nanocavity plasmonic modes under atomic-scale gap size changes. Therefore, the method gives access to the frequency-dependent radiative Purcell enhancement that a microscopic light emitter would undergo when placed at such nanocavity.

[1] IMDEA Nanociencia, Madrid, Spain. [2] Departamento de Física Teórica de la Materia Condensada and Condensed Matter Physics Center (IFIMAC), Universidad Autónoma de Madrid, Madrid, Spain. [3] Departamento de Física de la Materia Condensada and Condensed Matter Physics Center (IFIMAC), Universidad Autónoma de Madrid, Madrid, Spain. [4] Donostia International Physics Center (DIPC), E-20018 Donostia-San Sebastián, Spain. ✉email: fj.garcia@uam.es; roberto.otero@uam.es

The extreme confinement of the electromagnetic (EM) fields at the nanocavity[1] formed by a metallic nanostructure (nanoparticle or atomically sharp tip) and a metallic surface is essential for a number of emerging scientific and technological applications that exploit light–matter interactions at, and below, the nanoscale[2], such as ultrafast fluorescence imaging[3], single-molecule Raman spectroscopy[4], room-temperature quantum electrodynamics[5] or nanoscale metrology[6]. Light–matter coupling in these plasmonic nanocavities critically depends on the total photonic density of optical states (PhDOS) supported by the EM structure. This density of states has contributions from both radiative (usually termed "bright") and non-radiative (also coined as "dark") EM modes. When quantum emitters (organic molecules, quantum dots or monolayer materials) are placed within the gap region, the dynamics in their excitation decay is dictated by the total PhDOS of the nanocavity. In the weak-coupling limit, the population of the excited state displays an exponential decay with time, but the rate is different from that in vacuum, and the ratio between those rates is known as Purcell factor, which contains radiative contributions coming from the bright component of the PhDOS and non-radiative ones originated from dark modes[7]. In contrast, in the strong coupling regime, hybrid light–matter modes dubbed as polaritons are formed and the dynamics largely departs from being exponential, presenting the so-called Rabi oscillations in the population of the excited state of the emitter[8]. Due to their half-light character, polaritons inherit properties of both radiative and non-radiative EM modes. On the other hand, information of the total EM-field enhancement taking place in these systems[9,10] is usually extracted by analysing the radiation emerging from only the bright EM modes that, as commented above, is just one of the two contributions to the total EM field at the nanocavity. As a consequence, there is an increasing interest in the development of reliable experimental approaches able to disentangle the radiative and non-radiative contributions to the total PhDOS.

Single-molecule fluorescence techniques are able to quantify the Purcell enhancement experienced by dye molecules placed in the vicinity of metallic structures, and therefore allow measuring the total PhDOS[1,7], even with large spatial resolution[11]. However, they are unable to distinguish between the radiative and non-radiative contributions to the PhDOS. Recently, novel approaches exploiting photobleaching[12,13] experienced by fluorescent molecules have enabled discriminating the radiative and non-radiative Purcell enhancement mechanisms in plasmonic nanostructures. Nonetheless, they require a full statistical analysis performed on a large number of single-molecule experiments and operate within very narrow spectral windows, which restrict their widespread application in nanophotonics. Aside from optical techniques, the combination of electron energy loss[14,15] and cathodoluminescence[16] spectroscopies allows discriminating between radiative and non-radiative EM modes supported by metallic nanostructures[17,18], but its application to tip-on-surface nanocavities is hampered by the large scattering experienced by the probing electron beam when penetrating thick metallic regions, which precludes the incidence of the electron beam from the direction normal to the gap and, thereby, the excitation of gap modes, whose dipole moments are oriented perpendicular to the gap. It is thus apparent that new methods need to be developed to characterise the radiative contribution to the PhDOS for tip-on-surface nanocavitites.

A promising candidate is electroluminescence induced by a tunnelling current across a metallic gap[19–21], like that taking place in a Scanning Tunnelling Microscope (STM) junction[22–34], which has been recently used as the feed driving light emission by plasmonic nanoantennas[35]. STM luminescence (STML) spectra, however, carry information not only on the radiative EM modes of the nanocavity but also on the energy distribution of the tunnelling electrons[22,23,25–33]. Discriminating between optical and electronic effects is an unsolved issue that has limited the applicability of STML to the investigation of light–matter interaction phenomena in plasmonic nanocavities.

In this work we demonstrate that the radiative contribution to the PhDOS of tip-on-surface nanocavities can be obtained from STM experiments, by a new experimental framework that eliminates the electronic-structure factors from the STML spectra. Our procedure is based on the close relation between the inelastic current corresponding to an electronic energy loss $h\nu$ at a bias voltage $V_{\text{bias}}$, and the total tunnelling current at a different voltage $V_{\text{bias}} - \frac{h\nu}{e}$, which is measured by Scanning Tunnelling Spectroscopy (STS). In line with previous results[18,19,22–29,31], the peak positions and ratios in our raw STML spectra present a rather strong and complex dependence on the tunnelling parameters, which do not match the trends expected from EM calculations of the far-field power spectrum. Our new approach, however, yields spectra with constant peak ratios, and significantly lower shifts, which now agree with EM calculations within experimental error. We thus conclude that the combination of STML and STS makes STM a promising tool for the experimental characterisation of the radiative Purcell effect in tip-on-surface nanocavities, a capability with significant implications for the design and optimisation of light–matter coupling phenomena in plasmonic gaps.

## Results

**Dependence of STML spectra on the stabilisation bias.**
Figure 1a shows a sketch of our experiment. An electrochemically etched Au tip is brought to tunnelling distance $\delta$ from an atomically clean and crystalline Ag(111) surface, and tunnelling current $I_t$ is injected by the application of bias voltage $V_{\text{bias}}$. The inelastic part of this current can excite Localised Surface Plasmon Resonances (LSPR), some of which relax via the emission of photons of energy $h\nu$ that can be collected and analysed in the far-field. Figure 1b displays the evolution of the luminescence recorded with a tunnelling set-point of $I_t^{\text{st}} = 0.47$ nA for stabilisation voltages between 2.6 and 4.1 V at 4.5 K. The feedback loop is kept closed, so the tip-surface distance is different for each stabilisation voltage. Light spectra consist of a number of relatively broad peaks ($\sim$100 meV) with a general shape that depends on the specific geometry of the tip used for the experiment, and that can be controlled by tip modification procedures. Apart from relatively weak intensity modulations, all the spectra recorded for different tips show either one or two main peaks within an energy range from $\sim$1.5 to 3 eV. For the particular tip that corresponds to the STML spectra of Fig. 1, we find a high-intensity contribution at 2.53 eV, and a low-intensity peak at 3.0 eV, along with two weak shoulders at 2.3 and 2.0 eV.

For a given tip, the number of peaks and their corresponding intensities and energies depend on the tunnelling parameters (see Fig. 1b, c). The integrated light intensity increases rapidly with increasing stabilisation voltage up to about 3.4 V, after which it remains relatively unchanged (between 3.4 and 3.8 V in Fig. 1) and then decreases again for high enough voltages (above 3.8 V in Fig. 1). While the exact voltage ranges in which this evolution occurs depend on the particular tip used in the experiments, this overall trend is found for all the different tip configurations.

For relatively low stabilisation voltages (up to about 3.2 V in Fig. 1), the spectra show a cut-off at a maximum photon energy of $h\nu_{\text{co}} = eV_{\text{bias}}^{\text{st}}$, corresponding to the maximum energy that one electron can lose in an inelastic tunnelling process between the tip and the substrate (white oblique line in Fig. 1b, and vertical lines in Fig. 1c). Following previous literature[19,22,29], we will refer to this effect as the quantum cut-off. The transition to this regime is

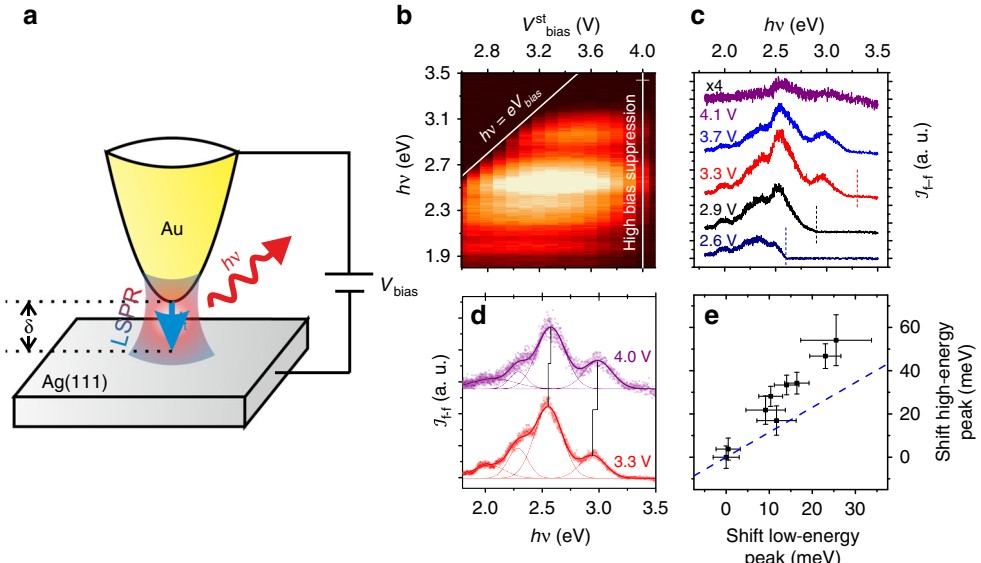

**Fig. 1 STM luminescence experiments. a** Sketch of the experiment. A gold tip is placed at a variable distance $\delta$ from a Ag(111) substrate and used in an STM to inject tunnel current $I_t$ with a bias voltage $V_{bias}$. The inelastic part of the tunnel current excites Localised Surface Plasmon Resonances (LSPR), and the radiative modes can decay by the emission of photons with energy $h\nu$ that are collected and analysed in the far field. **b** Light intensity emitted for a particular tip configuration as a function of the photon energy and the stabilisation voltage. Quantum cut-off and high-bias suppression are marked by white lines. **c** Individual spectra recorded for different stabilisation voltages. The quantum cutoff condition for each voltage (where relevant) is marked by a vertical line. The spectra have been vertically offset to enhance visibility, and the spectrum for a bias voltage of 4.1 V (above the high-bias cutoff) has been scaled to facilitate comparison. **d** Shift of the peak energies, extracted through the fitting to Gaussian lineshapes, with bias voltage (see discussion in Supplementary Note 1). **e** Proportionality between the energy shifts of the low- (dipolar plasmon) and high- (quadrupolar plasmon) spectral peaks. The blue dashed line corresponds to the expected trend from our EM calculations. Errors are estimated from the residual values of the fitting as the standard deviation of the fitting parameters.

rather smooth, being the light intensity significantly reduced at photon energies up to 100 meV below the bias voltage. For sufficiently high voltages (above 3.2 V in Fig. 1), tunnelling electrons have enough energy to excite all the localised plasmonic modes supported by the nanocavity, and the quantum cut-off is no longer relevant. The recorded far-field spectra are completely developed, but their exact shape (intensity ratios and widths) is still dependent on the stabilisation voltage (see for example the spectra in Fig. 1c). At even higher $V_{bias}^{st}$ (larger than 3.8 V in Fig. 1), a strong suppression of the overall intensity is observed (Fig. 1b, c). This high-bias intensity suppression has been previously reported for voltages at which tunnelling into bulk states on the noble metal surface or field-emission resonances leads to a strong increase in the tunnelling conductivity[29]. Tip-sample distance is thus enlarged under closed feedback conditions to maintain the tunnelling current constant. Importantly, the peak positions also change with different tunnelling parameters (see Fig. 1d). The low-energy contribution shifts to higher energy by about 25 meV when the stabilisation voltage is changed from 3.3 to 4.1 V, whereas the shift of the high-energy contribution is enlarged by a factor of 2.2 (about 55 meV, Fig. 1e). As we will discuss in the following section, this behaviour is not expected on the grounds of EM calculations, which predict a much similar shift for both contributions (blue line in Fig. 1d). It is worth noticing that modifying the stabilisation voltage under closed feedback conditions has the effect of changing the tip-surface distance and, thus, the optical response of the nanocavity. Tip-surface distances can be estimated from the conductivity at zero bias[36] (see Supplementary Note 2), yielding distances between 1 and 1.4 nm for bias voltages between 3.3 and 4.1 V. On the other hand, as mentioned in the introduction, STML spectra are also affected by the electronic properties of tip and sample. In order to distinguish between those two effects, it is worth comparing

STML spectra with the variations in the far-field light intensity as obtained by EM calculations.

**Comparison with theoretical calculations**. We model the STM tip as a gold sphere of radius $R$ separated a distance $\delta$ from the flat silver substrate. The excitation of plasmonic modes by inelastic tunnelling electrons in the nanocavity is described through a vertically oriented oscillatory electric point dipole source (with constant dipole strength), placed at the centre of the gap between sphere and surface. We have checked that the theoretical far-field spectra, calculated as the radiated power within the solid angle covered by the experimental detecting system, are rather insensitive to variations in the position of this dipole source. We anticipate that this is a consequence of the uniform field profile (capacitor-like) characterising low-order, radiative gap plasmonic modes. The total PhDOS is computed through the total radiated power flowing through a closed surface located within the gap of the nanocavity and containing the dipole source[37].

Figure 2a plots the theoretical far-field spectra, $\mathcal{I}_{f-f}$, for nanocavities with $\delta = 0.5$ nm and tip radii ranging from 1 to 20 nm. To facilitate the comparison among different spectra, the light intensity has been normalised to the square of the sphere radius, $R^2$, and a vertical offset has been introduced. We can observe that the spectra for small radii present two maxima, around 2.3 and 3.1 eV, that resemble with remarkable accuracy the experimental curves in Fig. 1. The character of these modes is revealed in Fig. 2c, d, which show maps for the resonant electric field amplitude, |**E**|, and induced surface charge distribution, $\rho$, evaluated at the two $\mathcal{I}_{f-f}$ peaks for $R = 5$ nm (green, see arrows). We can identify the two radiative modes responsible for the far-field maxima as dipolar (c) and quadrupolar (d) gap plasmons (note the uniform |**E**|-distribution between tip and

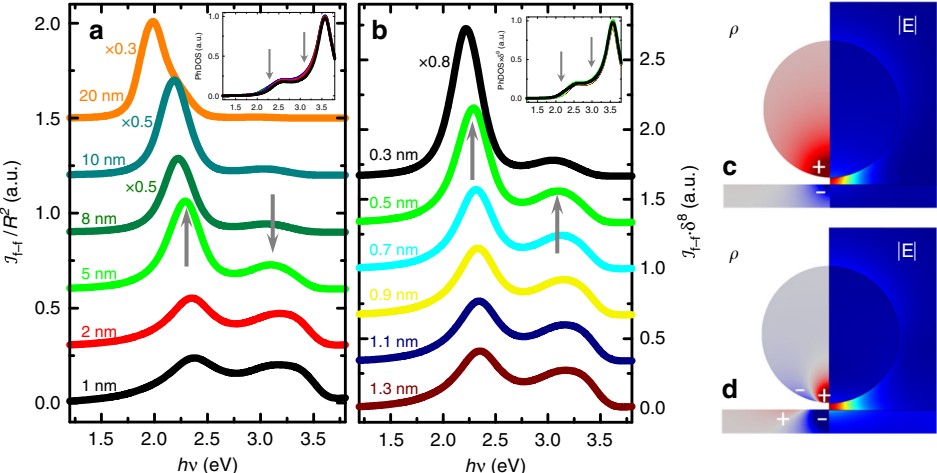

**Fig. 2 Modelling the tip-surface nanocavity.** Theoretical far-field (main panels) and PhDOS (insets) spectra for nanocavities comprising a gold sphere (tip) on top of a silver flat surface (substrate). Vertical dotted lines mark the peak maxima. **a** Spectra for different sphere radii, $R$, and 0.5 nm gap size, δ. **b** Spectra for different gap size, δ, and fixed sphere radius, $R = 5$ nm. The green curve plots the same spectrum in both panels. **c**, **d** render induced charge distribution (left) and electric field amplitude maps (right) for the two radiative plasmonic modes behind the far-field peaks indicated by grey arrows in **a**, **b**.

substrate). For larger tip sizes, only the low-energy (dipolar) peak is apparent, also in agreement with experimental data for other nanocavity samples (not considered here). Notice that this peak redshifts with larger $R$, in accordance with recent predictions in similar systems[38]. All these results are robust against subtle variations of tip geometry and are preserved for more realistic tip geometries in which the Au sphere is a nanoscale protrusion on a much flatter tip modelled as a wide-angle cone (see Supplementary Figs. 4 and 5 for further details).

The inset of Fig. 2a plots the total PhDOS for different values of $R$. In contrast to the far-field spectra, this magnitude is insensitive to variations in the tip radius, and present maxima at different photon energies. Whereas radiative plasmonic modes govern the light intensity spectra, strongly confined, higher energy resonances (which contribute to the so-called plasmonic pseudomodes) determine the near-field characteristics of metallic nanocavities[38,39]. These near-field resonances emerge as a result of the spectral overlapping of high-order dark plasmonic modes, and their spectral location is always in the vicinity of the metal surface plasmon frequency. Notice that, as explained in the introduction, these modes are a relevant part of the total PhDOS but are dark in the sense that they do not contribute to the radiation emerging from the structure. Thus, we can relate the PhDOS peaks to the plasmonic pseudomodes for the gold tip (2.5 eV) and silver surface (3.6 eV).

Figure 2b presents a similar study to Fig. 2a but for a fixed tip radius (5 nm) and tip-substrate distances ranging from 0.3 to 1.3 nm (in all cases the dipolar source is located at the gap centre). The inset shows the total PhDOS scaled by a factor $\delta^9$, showing that all the dependence of this magnitude on the gap size is given by this geometric factor. Note that the spectra for $\delta = 0.5$ nm and $R = 5$ nm (green curve) is the same in panels a and b. $\mathcal{I}_{f-f}$ spectra are multiplied by $\delta^8$ and shifted vertically to facilitate their comparison. All the curves in Fig. 2b exhibit the double-peak profile found in the experiments. The two maxima vary in opposite ways (the dipolar peak sharpens and increases while the quadrupolar one broadens and decreases) as the gap is reduced. Interestingly, however, both experience a very similar spectral shift. In Supplementary Fig. 4, a detailed analysis of the spectral shift of dipolar and quadrupolar peaks with δ is provided, revealing a linear trend with a proportionality ratio of about 1.15, significantly lower than that found in the experiments of the previous section (2.2 ± 0.1).

The comparison between experimental results and numerical EM calculations suggests that electronic-structure effects in the STML spectra play a critical role. In the following, we develop a fully experimental procedure to eliminate such effects from the STML spectra. We will demonstrate that this procedure does indeed fix all the problem of the shifts and, thus, allows for the direct experimental probing of the purely optical properties of the nanocavity.

**Relation between the excitation efficiency and the tunnel current.** If the nanocavity would support a single plasmonic mode of energy $h\nu$, the efficiency of excitation of such mode by the tunnelling electrons at a given stabilisation voltage would be proportional to the rate at which electrons can tunnel inelastically between occupied levels in one electrode (tip/substrate) and empty levels in the other one (substrate/tip), whose energy difference is equal to $h\nu$. One of these processes is depicted in Fig. 3a, where we assume for concreteness that the bias voltage is positive so that electrons flow from the tip to the substrate through the tunnel junction. Under these conditions, if the final state of the inelastic tunnel transition has energy $E$ (referenced to the Fermi level of the substrate), the energy of the initial state must be $E + h\nu - eV_{bias}$ (referenced to the Fermi level of the tip, see Fig. 3a). The total inelastic rate, $\mathcal{R}_{inel}$, will thus arise from the summation to all the possible inelastic processes compatible with such conditions (shaded region in Fig. 3a)

$$\mathcal{R}_{inel}(h\nu, V_{bias}) \sim \int_{-\infty}^{+\infty} \rho_T(E + h\nu - eV_{bias})f(E + h\nu - eV_{bias})\rho_S(E)$$
$$(1 - f(E))T_{inel}(E, h\nu, eV_{bias})dE,$$
(1)

where $\rho_T$ and $\rho_S$ are the electronic DOS of tip and sample, respectively, $f$ is the Fermi-Dirac distribution function and $T_{inel}$ is the inelastic transmission factor. Notice that $\mathcal{R}_{inel}$ will be negligible for photon energies that exceed $eV_{bias}$ by more than a few times $k_B T$. This situation corresponds to the quantum cut-off regime described above.

Next, we consider the hypothetical situation in which the applied voltage is $eV^*_{bias} = eV_{bias} - h\nu$ (Fig. 3b), while keeping the same tip-surface distance. In this situation, the initial and final states of the previous inelastic processes have the same energy

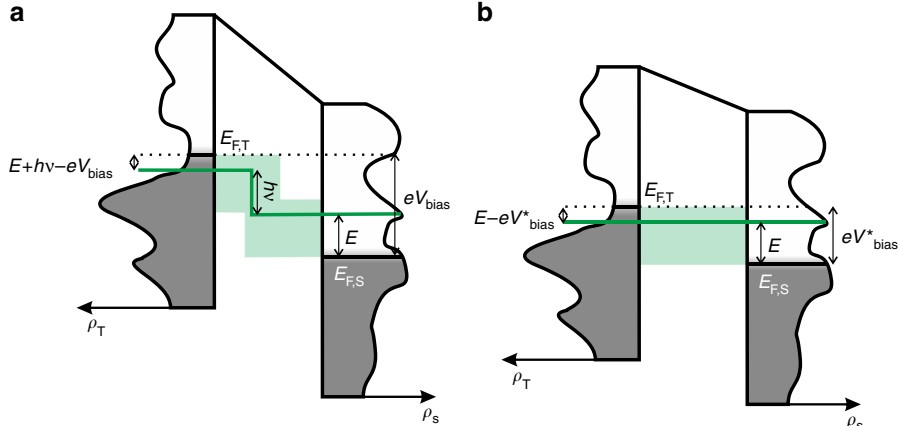

**Fig. 3 Relation between the rate of inelastic events and the tunnelling current. a** Sketch of an inelastic tunnelling event in which an electron crosses the tunnelling gap exciting a plasmon of energy $h\nu$ (green line). The light-green shaded region corresponds to all the inelastic processes that can contribute to plasmon emission for a given bias voltage. **b** Sketch of an elastic tunnelling event. The light-green shaded region corresponds to all the elastic processes that can contribute to the tunnelling current. Comparison of **a**, **b** shows that all the contributions to the inelastic channel correspond to contributions of the elastic channel for a lower bias voltage $eV_{bias}^* = eV_{bias} - h\nu$.

and can thus be coupled via elastic, instead of inelastic, tunnelling. As long as $eV_{bias}^*$ is larger than zero by more than a few times $k_B T$, the elastic tunnel current, $I_{el}$, at this voltage can now be calculated as

$$I_{el}\left(V_{bias}^*\right) \sim \int_{-\infty}^{+\infty} \rho_T\left(E - eV_{bias}^*\right) f\left(E - eV_{bias}^*\right) \rho_S(E) \tag{2}$$
$$(1 - f(E)) T_{el}\left(E, eV_{bias}^*\right) dE,$$

where $T_{el}$ is the elastic transmission function. Notice that all the factors in Eqs. (1) and (2) are the same if $eV_{bias}^* = eV_{bias} - h\nu$, except for the transmission factors. However, both elastic and inelastic transmission functions must depend upon the overlaps between the initial and final states, which are the same for $eV_{bias}^* = eV_{bias} - h\nu$ too. Thus, they are expected to depend exponentially on the energy (or, more precisely, on the root square of the energy). The inelastic transmission function could also depend on the electron and plasmon energies and on the bias voltage, but as long as this dependence is weaker than that of the overlap between initial and final states, we can safely assume that

$$T_{inel}(E, h\nu, eV_{bias}) \propto T_{el}(E, eV_{bias} - h\nu). \tag{3}$$

As we will see in the following, this hypothesis is completely fulfilled in our experimental data.

Based on the preceding considerations, we state that, for a single plasmonic mode of energy $h\nu$, the inelastic tunnelling rate for electrons that lose an energy $h\nu$ at a bias voltage $V_{bias}$ should be proportional to the elastic current at $eV_{bias} - h\nu$. Moreover, since the vast majority of the total tunnel current $I_t$ corresponds to elastic tunnel processes, the dependence of $\mathcal{R}_{inel}$ on the energy loss and positive bias voltage can be determined experimentally using

$$\mathcal{R}_{inel}(h\nu, V_{bias}) \sim \begin{cases} 0 & h\nu > e|V_{bias}| + O(k_B T) \\ \left|I_t\left(V_{bias} \mp \frac{h\nu}{e}\right)\right| & h\nu < e|V_{bias}| - O(k_B T), \end{cases} \tag{4}$$

where the upper and lower signs correspond to positive and negative voltages, respectively. This relation is illustrated in Fig. 4a. The rate of inelastic transitions decreases as the photon energy approaches the quantum cut-off condition since the total current decreases as the voltage drops to zero. Increasing the stabilisation voltage while keeping the feedback loop closed leads

to the increase of the tip-surface distance and, therefore to a decrease in the overall slope of the $I(V)$ curve. The modification of the stabilisation voltage also shifts the quantum cut-off in the inelastic transition rate.

**Combination of STML and STS measurements**. Plasmonic nanocavities, however, will show a continuous distribution of plasmonic modes at different energies, characterised by a density of optical states $\mathcal{D}(h\nu)$. Fermi's Golden Rule dictates that the inelastic current should result from the product of $\mathcal{D}(h\nu)$ and $\mathcal{R}_{inel}(h\nu, V_{bias})$. Notice that the expected enhanced probability for inelastic tunnelling in the presence of plasmonic modes is included in the factor $\mathcal{D}(h\nu)$, but it has no effect in the transmission functions since they are defined for individual tunnelling events with well-defined energies.

Finally, in order to stablish the link between the electronic and optical properties of the tunnel junction, it has to be noticed that not all the plasmonic modes show similar dipole moments and, thus, their radiative power is different. Here we emphasise that the density of states associated with dark modes cannot contribute to the luminescence. Thus, the far-field light intensity at $h\nu$ in a tunnel junction biased by $V_{bias}$, $\mathcal{I}_{f-f}(h\nu, V_{bias})$, will be determined by the radiative density of optical states, characterised by the radiative power of the nanocavity $P(h\nu)$, as follows

$$\mathcal{I}_{f-f}(h\nu, V_{bias}) = P(h\nu)\mathcal{R}_{inel}(h\nu, V_{bias}) \sim P(h\nu)I_t(eV_{bias} - h\nu), \tag{5}$$

where the second identity is valid for $h\nu < eV_{bias} - O(k_B T)$, according to Eq. (4). Thus, the bare STML spectra can be decomposed into two factors. The first one accounts exclusively for the optical properties of the nanocavity, while the second one includes all the electronic properties of the junction, along with the probability for plasmon excitation.

The considerations above suggest a fully experimental normalisation procedure to extract solely the radiative optical response of the nanocavity. First, for a given set of stabilisation tunnelling parameters ($I_t^{st}$ and $V_{bias}^{st}$), we record both the far-field light intensity and the $I(V)$ curve. Second, we normalise the light intensity at each photon energy by the tunnel current at the voltage $eV_{bias}^{st} - h\nu$ (see Fig. 4b–d). Applying this procedure to all the datasets in Fig. 1, we obtain the spectra in Fig. 5a, b. Figure 5a shows the normalised far-field light intensity in colour scale as a

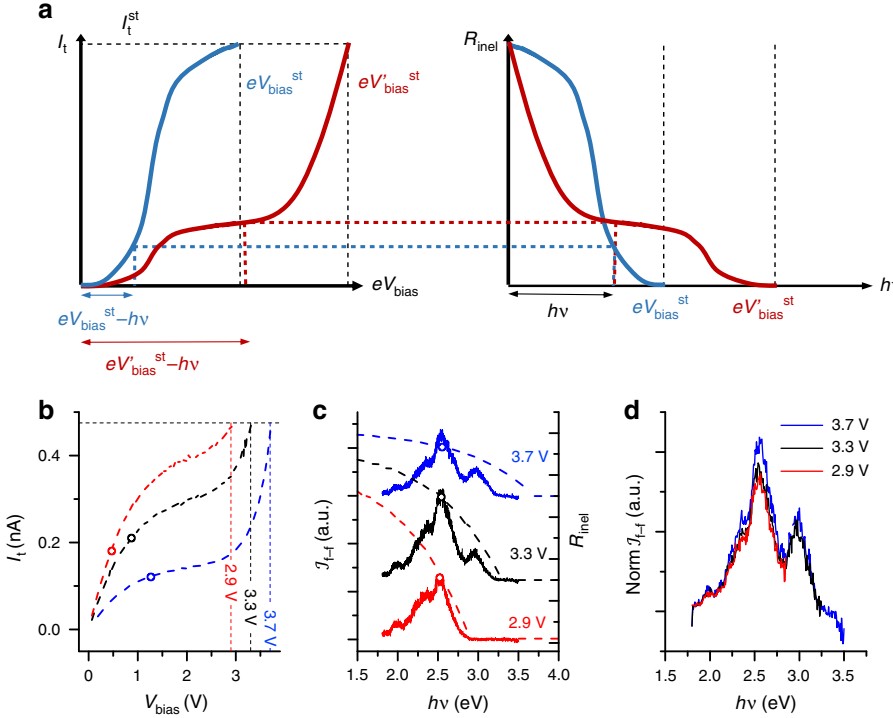

**Fig. 4 Example of the normalisation procedure. a** Schematic representation of the relationship between the inelastic rate and the tunnelling current in Eq. (4) for two values of the stabilisation bias voltage. The dashed-colour lines represent the contribution to the inelastic rate to the same photon energy for both voltages obtained from the $I(V)$ curves. **b** Experimental $I(V)$ curves recorded with $I_t^{st} = 0.47$ nA and bias voltages of 2.9, 3.3 and 3.7 V, respectively, measured right before the corresponding luminescence spectra. **c** Luminescence spectra (solid lines) recorded with the same tunnelling parameters as in **b** and the rate of inelastic events (dashed lines) as given by Eq. (4). The solid **c**ircles in **b** and **c** correspond to the tunnelling intensities (and thus inelastic rates) at the voltages that correspond to the photon energy of 2.55 eV (dipolar contribution). **d** Normalised luminescence spectra.

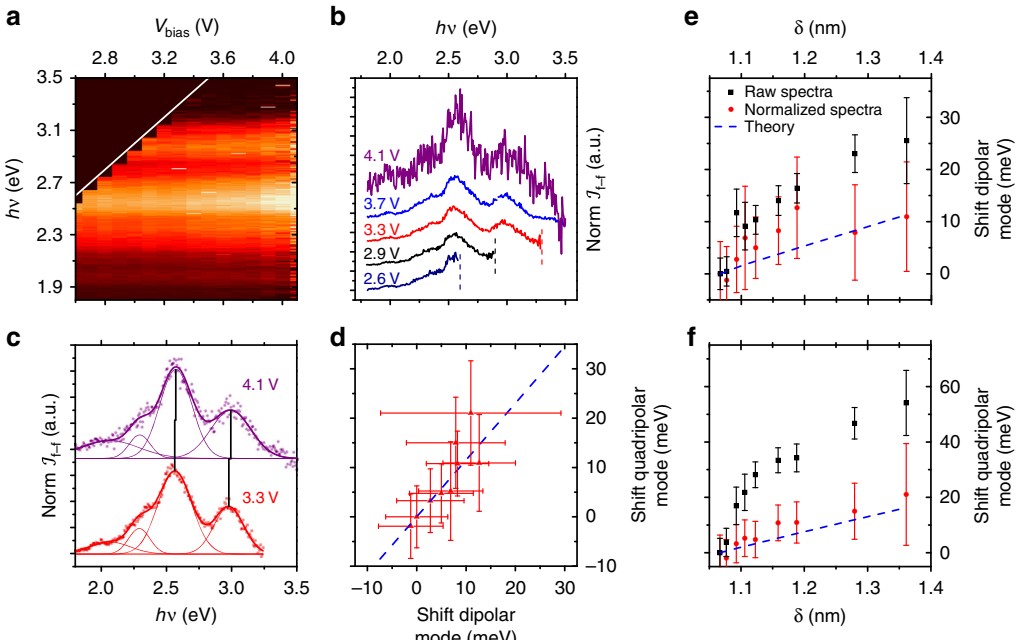

**Fig. 5 Normalised STM spectra and the shifts of the plasmonic resonances. a** Normalised light intensity emitted for a particular tip configuration as a function of the photon energy (vertical axis) and the stabilisation voltage (horizontal axis). Quantum cut-off is marked by a white line. **b** Individual spectra recorded for different stabilisation voltages. The quantum cutoff condition for each voltage (where relevant) is marked by a vertical line of the same colour as the spectra. The spectra have been vertically offset to enhance visibility. **c** Shift of the peak energies with bias voltage. The peak position has been extracted by fitting the spectra to Gaussian peak shapes. **d** Proportionality between the energy shifts of the low- and high-energy peaks. The blue dashed line corresponds to the expected trend from our EM calculations. **e, f** Dipolar and quadrupolar mode shifts obtained from the raw spectra (black squares), the normalised spectra (red circles) and EM calculations (blue line) for gap sizes (tip-surface distances) estimated from the zero-bias conductance at different stabilisation voltages. Error bars in **d–f** correspond to the standard deviation from the Gaussian fittings.

function of the photon energy (*x*-axis) and the stabilisation bias (*y*-axis) for every curve in Fig. 1. Notice that most of the dependence of $\mathcal{I}_{f-f}$ on the bias voltage has been removed. Similar results can be obtained for different tips and at negative voltages, where the role of the electronic surface state of Ag(111) can be discriminated (see Supplementary Note 7 and Supplementary Fig. 6). However, with respect to the optical characterisation of the nanocavity, the most relevant effect is that the shifts of the peaks are now in very good agreement with the EM calculations. Figure 5c shows that, after normalisation, the shift of the peaks for $V_{bias}^{st}$ between 3.3 and 4.1 V is significantly lower than that found for the raw STML spectra in Fig. 1 and much closer to each other than before normalisation (about 15 meV for the dipolar mode and 20 for the quadrupolar mode). This ratio between the shifts of the high- and low-energy modes corresponds to the theoretical expectation within experimental accuracy (Fig. 5d). Moreover, when plotted as a function of the gap size estimated from the zero-bias conductance, the shifts of the low- and high-energy peaks match with the calculated shifts of the dipolar and quadrupolar modes for such tip-surface distances (Fig. 5e, f).

## Discussion

The results shown above demonstrate that our normalisation procedure effectively removes the electronic factor from STML spectra and, thus, allows for a direct inspection of the radiative plasmonic modes supported by the tip-sample nanocavity. In particular, the integrated intensities of the normalised spectra only show a weak monotonic increase with stabilisation voltage, instead of the non-trivial dependence reported in Fig. 1b. Interestingly, the quantum cut-off is much more abrupt in the normalised spectra compared to the raw ones, the peak ratios are almost constant and the high-bias suppression is eliminated (an extended discussion on these issues can be found at the Supplementary Notes 4–6). Moreover, the agreement of the shifts in the normalised spectra with EM calculations demonstrate that the luminescence peaks in raw STML can be shifted up to several tens of meV with respect to the far-field optical spectrum since they arise from the product of the latter and the tunnelling current, which has a non-negligible and non-constant slope. Thus, the procedure described here to eliminate such electronic-structure factors from the STML spectra provides a unique tool to investigate the radiative plasmonic modes of tuneable nanocavities with both meV spectral and sub-nanometric spatial resolutions.

To conclude, we have demonstrated that the radiative modes of a plasmonic nanocavity can be studied by a combination of STML and STS through a novel procedure that eliminates all the electronic-structure contributions to the measured far-field optical spectra. The method is based on the relationship between the rate of inelastic tunnelling events with energy loss $h\nu$ at a bias voltage $V_{bias}$ and the total tunnel current at a lower bias $eV_{bias} - h\nu$. While our set of raw spectra show a rather strong and non-trivial dependence with the bias voltage, after normalisation this dependence is removed. The comparison against theoretical calculations allows us to link our experimental findings with the radiative characteristics of the plasmonic modes supported by sub-nanometric gaps. By using this new technique, we have been able to study in depth the evolution of the spectral locations of the dipolar and quadrupolar plasmonic modes as a function of the gap size with a meV frequency resolution. Our findings reveal STM as an essential tool for the optical characterisation of plasmonic nanocavities, as well as light–matter interaction phenomena taking place at their gaps.

## Methods

**Sample and tip preparation**. The experiments were performed with an Omicron Low-Temperature Scanning Tunnelling Microscope (LT-STM), operated at 4.5 K, and in Ultra-High-Vacuum (UHV) conditions ($P \sim 10^{-11}$ mbar). Clean Ag(111) samples were prepared by repeated cycles of sputtering with 1.5 keV Ar$^+$ ions for 10 min, followed by 10 min of thermal annealing at 500 K. To enhance the plasmonic response of the tunnel junction the tips were made of Au. The Au tips were electrochemically etched in a solution of HCl (37%) in ethanol, at equal parts, and cleaned in UHV by sputtering with 1.5 keV Ar$^+$ ions for 50 min.

**Light collection**. To collect the emitted light we modified the head of our STM following the procedure described in ref. [40]. Our light detection set-up is formed by three lenses, one in UHV and two in air, three mirrors and an optical spectrometer (Andor Shamrock 500) equipped with a Peltier cooled Charge-Coupled-Device (Newton EMCCD). The first lens collimates the photons from the tunnel junction outside the UHV environment through a BK7 viewport. It is a plane-convex lens placed 30 ± 5 mm away from the centre of the sample stage, forming an angle of 70° with respect to the long axis of the tip. It has a numerical aperture of NA = 0.34, and a solid angle of collection of 0.26 sr. Once the emitted photons are outside the UHV chamber, they are guided by three plane mirrors and two more BK7 plane-convex lenses placed on top of a pneumatic table to isolate the system from mechanical vibrations. The two lenses (lens 2 and 3) have focal lengths of 300 and 200 mm, respectively. Finally, the beam of light enters the optical spectrometer (Andor Shamrock 500), that has three interchangeable gratings with groove densities of 150, 300 and 1200 l/mm, with band-passes of 331, 163 and 38 nm, respectively. The first grating has the blaze at 300 nm, and the other two at 500 nm. The detection of light is performed with a Peltier cooled Electron-Multiplying Charge-Coupled-Device (Newton EMCCD) at the end of the spectrometer. The sensor is back-illuminated and has 1600 × 400 pixels, with an area of 16 µm/pixel. All experiments were done with the EMCCD at −85 °C. The presented spectra are not corrected by the efficiency of the set-up.

**Theoretical calculations**. The numerical simulations were carried out using the frequency-domain finite element solver of Maxwell's Equations implemented in the commercial software Comsol Multiphysics. We have exploited the cylindrical symmetry of the theoretical geometry: a spherical gold tip on top of a flat silver surface driven by an oscillating electric dipole located within the gap and parallel to the symmetry axis. Thus, we have performed radiative and nonradiative Purcell enhancement calculations using a 2D axial-symmetric model. This decreased sensibly the computational times and allowed for a thorough analysis of the system considering large ~30λ simulation domains (excluding perfect matching layers). A conformal mesh distribution was employed to describe EM field propagation from the (sub-)nanometric cavity gap, driven by a point-like dipole placed at the gap centre, to the far-field detector, where Poynting vector was integrated. The convergence of numerical results against the mesh size and distribution was checked. The permittivity of gold and silver were taken for the experimental fittings provided in ref. [41].

## Data availability

The data that support the findings of this study are available from the corresponding author upon reasonable request.

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

## Acknowledgements

We acknowledge financial support from the Spanish Ministry for Economy and Competitiveness (grants FIS2015-72482-EXP, FIS2015-64951-R, FIS2016-78591-C3-1-R, PGC2018-098613—B-C21, PGC2018-096047-B-I00, RTI2018-099737-B-I00 and MAT2014-53432-C5-5-R), the regional government of Comunidad de Madrid (grant S2018/NMT-4321), Universidad Autónoma de Madrid (UAM/48 and UAM/134) and IMDEA Nanoscience. Both IMDEA Nanoscience and IFIMAC acknowledge support from the Severo Ochoa and Maria de Maeztu Programmes for Centres and Units of Excellence in R&D (MINECO, Grants SEV-2016-0686 and MDM-2014-0377). We also acknowledge support by the QuantERA program of the European Union with funding by the Spanish AEI through project PCI2018-093145.

## Author contributions

A.M-J. and K.L. performed the STML experiments described in this paper. A.I.F-D. carried out the EM calculations. The construction and optimisation of the experimental set-up was performed by A.M-J., K.L. and D.G. Experimental data were analysed by A.M-J. and R.O. The writing of the paper were largely in charge of F.J.G-V., R.M. and R.O.

## Competing interests

The authors declare no competing interests.
