## [Peer Review File · Nature Communications]

Reviewers' Comments:

Reviewer #1:

Remarks to the Author:

This is a review on "Unveiling the Radiative Local Density of Optical States of a Plasmonic Nanocavity by STM Luminescence and Spectroscopy" by Martín-Jiménez et al.

This manuscript presents a rationale for an experimental study of electroluminescence from plasmonic nanogaps. Luminescence induced from electron tunneling usually contains correlated information from both the local density of radiative photonic states, the observable of interest in the characterization of plasmonic systems, and from the metal's electronic bandstructure (electronic DOS). The resulting luminescence spectra show relative peak amplitude and spectral position that do not correspond to expectations from electromagnetic theory because of the electronic DOS contribution. This manuscript reports on a new experimental procedure, combining STM luminescence (STML) and STM spectroscopy (STS), that eliminates the latter, providing information solely on the optical properties of the nanocavities, and in keeping with expectations from EM simulations.

The Authors test the proposed procedure studying the bias-dependent luminescence from a gap formed between a Ag(111) surface and a Au STM tip. The luminescence shows several spectral features which spectral position and amplitude are bias-dependent, in ways which are compatible with unwanted electronic contributions. This is confirmed by photonic DOS calculations done with FEM simulations for varying tip radii and tip-sample distance. The Authors discuss and rationalize the origin of the electronic contribution and demonstrate that it can be removed by normalizing the far-field electroluminescence by the tunneling current at each bias.

This is an excellent report providing an ingeniously simple solution to the suitable study of plasmonic nano cavities using electroluminescence. The proposed method is robust and both the experimental results and the theoretical rationalization of the method are convincing and adequate. I think the manuscript is clearly written overall. This includes the SI as well as the methods section. I would simply suggest the following:

- 1) Tip radius variable R should be introduced in the text on line 142.
- 2) It is unclear to me whether the FEM is a 2D model with a magnetic point dipole pointing out of the plane (all electric field orientations driven inside the plane of the structure), or a reduced 3D model, in which case I am not sure in which direction the point-dipole is oriented. Additional details would clarify the matter for the reader.

Reviewer #2:

Remarks to the Author:

The present work reports an experimental normalization procedure to eliminate the influence of the electronic structure of a given tip-sample system in STM to the electroluminescence spectra. According to comparison with electromagnetic simulations, the corrected STML spectra provide 'clean' access to the radiative plasmonic modes of the junction/nanocavity. This is novel and an interesting aspect for the STML community.

The drawn conclusions are based on the (experimental and simulated) dependence of the far-field/STML spectra on the tip-sample distance, in particular on the energetic shifts of the radiative plasmon modes with gap distance. The conclusions are, however, based on assumptions which need better justification. The main conclusion of the manuscript relies on comparison of experimental data with simulated results based on different geometries. In order to provide strong evidence for the validity of the conclusions, the authors need to address the respective points raised below.

The figures and data are of good quality and described properly throughout the manuscript. The

descriptions of experimental and theoretical procedures are clear and complete and allow for reproducibility of the work by other researchers.

The presented normalization procedure is an interesting recipe for future STML studies and provides a deeper understanding of the signatures in STML spectra and their connection to the radiative modes in plasmonic STM junctions. However, the work does not provide fundamentally new insight into the properties of plasmonic nanocavities, and wide-ranging implications for future research are not sufficiently discussed. The procedure is mainly relevant for the STML community, but its novelty is limited in view of new physical insight. In order to meet the criteria of Nat. Comm. the authors need to point out more clearly the relevance of their work for nanophotonic applications and the current understanding of light-matter interaction in plasmonic systems.

In that regard, I cannot recommend publication of the current version of the manuscript in Nature Communication.

Specific comments are provided in the following:

1. The introduction is poorly written, not well structured and needs significant revision. A clearer and well-structured introduction into the research field, the basic concepts and terms, especially for non-expert readers, is required.
 - In particular, proper introduction to the essential photonic properties of plasmonic (lossy) nanocavities regarding pDOS, radiative/non-radiative (dark and bright) plasmonic modes, Purcell enhancement (radiative and non-radiative/losses) and the connections between these terms is missing, ideally supported with some formulas. A short comprehensive overview of the current understanding and open questions should be given.
 - What are dark and bright plasmon modes, what are their characteristics, what are they relevant for, which ones are probed here?
 - It is clear that STML can probe the radiative plasmon modes (using the normalization procedure). But it is barely explained how this connects to the pDOS. How does STML access the pDOS (as suggested in the title)? Why is STML a good probe? How does this compare to other methods, and what are the advantages/disadvantages? What is the novel insight using STML? These things are partially contained in the introduction, but are missing well order and conclusive explanations. All these points should become clear immediately also to non-expert readers without the need to do major literature research. This is not fulfilled in the current version of the manuscript.
2. The manuscript has some grammatical/expressional errors, for example:
 - line 18: shouldn't it say "... plasmonic modes to the photonic density of states..." ?
 - line 37: "... has focused on the spectral characterization..."
 - line 87: "...main peaks within an energy range from..." (without comma)
 - line 99: "...cutoff at a/the maximum photon energy..."
 - line 153: "...As a difference with..." is not a good expression for a comparison. It should say something like "Compared to...", or "in contrast to...".
3. The description in line 41 is not unambiguous: "...able to map the radiative...". Map spatially, spectrally? This needs more detailed discussion.
4. Some references need revision and more careful literature research, for example
 - line 35: there has been a lot newer work published since 2013, with great achievements and high novelty. Also, the list in line 35 misses novel work on strong coupling in plasmonic cavities, which is directly relevant for the submitted work.
 - line 49: Some relevant references to EELS studies are missing (e.g. <https://doi.org/10.1021/nn900922z>, or <https://doi.org/10.1021/nl5043775>)
 - line 37: this sentence needs a reference (what techniques, what are the results?)
5. The authors claim that EELS is problematic for purely metallic samples because of large scattering of the electron beam. But in Ref. 13 and other work, metallic samples are investigated,

partially also combined with CL (e.g. Losquin et al. 2015). Could the authors provide some evidence/reference for the stated problem of strong scattering, and what this implies?

6. A concept image/experimental scheme in Figure 1 would be nice and helpful.

7. How robust is the fitting of the peaks in the STML spectra, especially regarding the peak position of the 2.6 eV peak? The low-energy wing of the spectra below 2.5 eV, as for example that at 3.3 V in Fig. 1 c), seems very structured (with potentially more than two peaks underneath), and it is questionable if this can be fitted unambiguously by two gaussians. How sensitive is the factor 2.2 to the fitting? As the relevant shifts are in the few 10 meV range, robustness of the fit needs to be carefully discussed.

8. The authors model the tip-sample nanocavity by a sphere on a flat substrate. The STM tip, however, is not a closed nanoparticle but a half-terminated structure, which might support very different plasmonic modes. Moreover, STM tips often exhibit small nanoprotusions on a larger tip apex due to tip conditioning, which deviates strongly from the employed sphere model. How much are the results dependent on the geometrical shape of the nanoparticle? Would e.g. a nanorod or conical nano-cylinder yield similar results? How much do the gap modes depend on the semi-infinite tip shaft? The authors should justify the use of a sphere model and provide evidence that this yields comparable and correct plasmonic modes.

9. Related to 7: The simulations yield a proportionality factor of 1.15 between the spectral shift of the dipolar and quadrupolar modes. How does this factor depend on the geometry (e.g. sphere vs. nanorod)? Does this provide a valid (universal) reference for the tip geometry? As the main conclusion of the manuscript relies on the comparison of the experimentally observed shifts to that number, this needs to be clarified.

10. (How) do the number of peaks and their spectral distribution vary for different tips? Given that the manuscript intends to provide a universal 'recipe' how to use STML to probe radiative plasmon modes in nanocavities, its sensitivity on the exact geometry and tip condition should be discussed, and reproducibility of the procedure for at least a second tip needs to be shown.

11. The title suggests experimental probing of the radiative phDOS, but it is poorly discussed in the manuscript how the STML spectra are connected to the phDOS. The authors should explain more clear how STML accesses the (radiative) phDOS.

12. The authors introduce plasmonic pseudomodes to explain the peaks in the simulated phDOS (starting line 153). Whereas the comparison of STML spectra to the simulated far-field spectra is obvious and clear, the connection to the phDOS and the role of the pseudomodes for SMTL remains vague. A clear connection of the pseudomodes and phDOS to the STML/far-field spectra and a detailed discussion of its relevance for the experimental data should be given.

13. An explanation of the gray errors in Fig. 2 should be given in the figure caption.

14. The phDOS (far-field spectra) in Fig. 2b) are scaled by δ^9 (δ^8). Is there a physical meaning of these exponents? Or is this used based on empirical findings? Such numbers should be explained properly.

15. The authors claim that the dependence of the inelastic transmission function on the photon energy can be neglected. However, one could expect that the probability to excite a plasmon via a tunneling electron depends on the available photonic density of states of the nanocavity, which is a function of the photon energy. In that sense, the authors need to justify this assumption, and it needs to be explained why this dependence is (significantly) weaker than the overlap of initial and final states.

16. At the end of the results section it has been explained that the STML spectra after normalization are governed solely by the radiative plasmonic modes in the nanocavity. Again, it however remains unclear to the reader how this connects to and unveils the radiative phDOS, as suggested in the title. In that regard the title is not chosen adequate enough.

17. The discussion part would benefit from a more detailed discussion on the wide-ranging implications of the results for the understanding of the photonic and plasmonic properties of nanocavities (also compared to other methods), beyond the technical achievement to use STML as a probe for the radiative properties of such nanocavities.

Reviewer #3:

Remarks to the Author:

In this manuscript, the authors report an experimental procedure to evaluate optical response of a plasmonic nanocavity formed between an STM tip and metal substrate. They showed bias voltage dependency of STML spectrum and explained the main spectral features with the aid of theoretical EM simulations. Then they discussed the influence of electronic structures of the STM tip and metal substrate on STML spectra to derive a simple expression for rate of inelastic tunneling (4). Finally, they demonstrated that STML spectra can be very easily normalized to show only optical properties of the plasmonic nanocavity which is independent of the electronic structures of the STM junction.

The major novelty of the present manuscript lies in establishment of an easy method to disentangle the optical and electronic properties of a nanocavity, which can be utilized widely in this kind of experiment. The demonstration is remarkable to my perspective since the extremely small size of the electromagnetic field at the nanocavity usually makes it very difficult to understand the nature of the field itself, and, at the same time, the small EM field is the source of many intriguing application of plasmonics. For deeper understanding of nanocavity plasmons STM combined with an optical system is a promising platform. With the achievement in this manuscript I expect we start to understand the nature of the important EM field.

I recommend publication of this article in Nature Communications after my criticism listed below has been taken into account.

1. I believe the most important part of this study is the normalization procedure, so I recommend to put Figures S1 and S2 into the main text.

2. To generalize the findings, several different data sets (different tip conditions) should be added. This also contributes to making the effectiveness of the method clearer.

3. In line with the previous comment, raw experimental data and normalized data measured at both negative and positive voltages should be added and discussed.

4. The authors suggested 5 nm tip radius based on the EM calculations. However, 5 nm sound too small for a radius of an STM tip and I believe the real STM tip radius is much larger than 5 nm. If it is possible, addition of an SEM image showing the tip radius is desirable, or adds some comments on discrepancy between the experiment and theoretical simulation.

5. It is a well-known fact that Ag(111) surface has a surface electronic state located around 50 meV below the Fermi level, and it can be expected that the surface state play a role in the inelastic tunneling process based on the conclusions of this work. I believe that the quality of this work would be considerably improved if the authors can show the signature of the surface state in STML spectrum.

Minor comments;

1. The first sentence in the abstract is not easy to understand, especially to general people.

Probably because too much jargons are used.

2. I recommend to include a schematic diagram to Fig.1 which illustrate the experiment in a simple way.

3. The optical system consists of three plano-convex lenses to lead the emitted light from a point underneath the STM tip to the detector. This is not a normal setup, because even number of plano-convex lenses should be used. It is helpful to includes a schematic diagram of the experimental setup.

4. I don't understand why two plano-convex lenses outside of the camber have very long focal lengths, 300 and 200 mm.

5. In the expression (S2), z should be δ .

6. In figure S3 (b) the vertical axis should be angstrom.

**Subject: Reply to referees –
“Unveiling the Radiative Local Density of Optical States of a Plasmonic
Nanocavity by STM Luminescence and Spectroscopy”**

Dear Referees,

We would like to thank you for your effort in refereeing our manuscript. Indeed, your referee reports have been extremely helpful to identify a few unclear points in the original version of the manuscript, and your constructive criticisms have sparked new experiments, analysis and calculations that have also helped us to build a more robust case for our conclusions. Including all this new work, we have generated a new version of the manuscript which we believe meets every demand you made in your reports. In the following, we discuss all the points you raised, and describe the changes performed on the manuscript to address them:

Reviewer #1 (Remarks to the Author):

This is a review on “Unveiling the Radiative Local Density of Optical States of a Plasmonic Nanocavity by STM Luminescence and Spectroscopy” by Martín-Jiménez et al.

This manuscript presents a rationale for an experimental study of electroluminescence from plasmonic nanogaps. Luminescence induced from electron tunneling usually contains correlated information from both the local density of radiative photonic states, the observable of interest in the characterization of plasmonic systems, and from the metal’s electronic bandstructure (electronic DOS). The resulting luminescence spectra show relative peak amplitude and spectral position that do not correspond to expectations from electromagnetic theory because of the electronic DOS contribution. This manuscript reports on a new experimental procedure, combining STM luminescence (STML) and STM spectroscopy (STS), that eliminates the latter, providing information solely on the optical properties of the nanocavities, and in keeping with expectations from EM simulations.

The Authors test the proposed procedure studying the bias-dependent luminescence from a gap formed between a Ag(111) surface and a Au STM tip. The luminescence shows several spectral features which spectral position and amplitude are bias-dependent, in ways which are compatible with unwanted electronic contributions. This is confirmed by photonic DOS calculations done with FEM simulations for varying tip radii and tip-sample distance. The Authors discuss and rationalize the origin of the electronic contribution and demonstrate that it can be removed by normalizing the far-field electroluminescence by the tunneling current at each bias.

This is an excellent report providing an ingeniously simple solution to the suitable study of plasmonic nano cavities using electroluminescence. The proposed method is robust and both the experimental results and the theoretical rationalization of the method are convincing and adequate. I think the manuscript is clearly written overall. This includes the SI as well as the methods section. I would simply suggest the following:

- 1) Tip radius variable R should be introduced in the text on line 142.*
- 2) It is unclear to me whether the FEM is a 2D model with a magnetic point dipole pointing out of the plane (all electric field orientations driven inside the plane of the structure), or a reduced*

3D model, in which case I am not sure in which direction the point-dipole is oriented. Additional details would clarify the matter for the reader

We thank Referee 1 for his/her enthusiastic report. We have addressed both suggestions as follows:

- 1) The meaning of R as the tip radius variable is now explicitly mentioned in line 160 of the main text.
- 2) We have now clarified the electric nature of the dipole sources and the axial-symmetric character of the simulation domains in lines 149-150 in the main text and lines 351-352 in the Methods section.

Reviewer #2 (Remarks to the Author):

The present work reports an experimental normalization procedure to eliminate the influence of the electronic structure of a given tip-sample system in STM to the electroluminescence spectra. According to comparison with electromagnetic simulations, the corrected STML spectra provide 'clean' access to the radiative plasmonic modes of the junction/nanocavity. This is novel and an interesting aspect for the STML community.

The drawn conclusions are based on the (experimental and simulated) dependence of the far-field/STML spectra on the tip-sample distance, in particular on the energetic shifts of the radiative plasmon modes with gap distance. The conclusions are, however, based on assumptions which need better justification. The main conclusion of the manuscript relies on comparison of experimental data with simulated results based on different geometries. In order to provide strong evidence for the validity of the conclusions, the authors need to address the respective points raised below.

The figures and data are of good quality and described properly throughout the manuscript. The descriptions of experimental and theoretical procedures are clear and complete and allow for reproducibility of the work by other researchers.

The presented normalization procedure is an interesting recipe for future STML studies and provides a deeper understanding of the signatures in STML spectra and their connection to the radiative modes in plasmonic STM junctions. However, the work does not provide fundamentally new insight into the properties of plasmonic nanocavities, and wide-ranging implications for future research are not sufficiently discussed. The procedure is mainly relevant for the STML community, but its novelty is limited in view of new physical insight. In order to meet the criteria of Nat. Comm. the authors need to point out more clearly the relevance of their work for nanophotonic applications and the current understanding of light-matter interaction in plasmonic systems.

In that regard, I cannot recommend publication of the current version of the manuscript in Nature Communication.

We thank Referee 2 for his/her positive opinion about the quality of the work and the relevance of our research for the STML community. We understand that, in the previous version of the manuscript, the broader implications of our work were not sufficiently stressed and/or clarified. We have now completely rewritten the introduction to underline the main aspect of our work: our experimental technique allows gaining access to the

radiative electromagnetic modes of the plasmonic nanocavity between a metallic tip and a metallic surface. This type of nanocavities are relevant for many different applications, now mentioned in the introduction, but the study of their optical properties has been somewhat hampered by their characteristic geometry. Thus, different optical methods have been successfully exploited to study the total Photonic DOS (PhDOS), but they do not allow disentangling the contribution from radiative modes, the ones that would govern the far-field signatures of the polaritonic states that could potentially be formed when placing a quantum emitter at the nanocavity. On the other hand, cathodoluminescence (CL) microscopy allows, in principle, characterizing the radiative modes of plasmonic nanocavities. However, for this particular geometry, in which the dipoles oscillate along the direction perpendicular to the surface (along the axis of the tip), the excitation of these plasmonic modes would require a normal incidence of the electron beam, which is precluded due to the shadowing effect of the tip. In a sense, thus, we can regard our method as a way to launch electrons in the perpendicular direction of the surface, by making them flow through the metallic tip onto the surface via tunneling.

Specific comments are provided in the following:

1. The introduction is poorly written, not well structured and needs significant revision. A clearer and well-structured introduction into the research field, the basic concepts and terms, especially for non-expert readers, is required.

- In particular, proper introduction to the essential photonic properties of plasmonic (lossy) nanocavities regarding phDOS, radiative/non-radiative (dark and bright) plasmonic modes, Purcell enhancement (radiative and non-radiative/losses) and the connections between these terms is missing, ideally supported with some formulars. A short comprehensive overview of the current understanding and open questions should be given.

- What are dark and bright plasmon modes, what are their characteristics, what are they relevant for, which ones are probed here?

- It is clear that STML can probe the radiative plasmon modes (using the normalization procedure). But it is barely explained how this connects to the phDOS. How does STML access the phDOS (as suggested in the title)? Why is STML a good probe? How does this compare to other methods, and what are the advantages/disadvantages? What is the novel insight using STML? These things are partially contained in the introduction, but are missing well order and conclusive explanations.

All these points should become clear immediately also to non-expert readers without the need to do major literature research. This is not fulfilled in the current version of the manuscript.

As mentioned earlier, the introduction has been completely rewritten to clarify all these points. In particular, the nature of bright and dark plasmonic modes is now described in detail in lines 39-41 and the so-called Purcell enhancement is described in lines 43-47 in connection to the plasmonic modes introduced above. We have also tried to clarify our statements somewhat further. For example, we do not claim that STML gives access to the total PhDOS, which on the other hand would not be so novel, since, as discussed above, optical techniques have been previously used to that effect. Our claim, instead, is that STML gives access to the radiative contribution to the PhDOS, which is characterized by the radiated power of the nanocavity. This is now further stressed in the two paragraphs from line 249 to 262, where we discuss the relation between the tunnel current and the far-field intensity emerging from our analysis. In this respect, the characterization of the radiative modes sustained in this kind of tip-on-surface plasmonic nanocavities by STML is, to our knowledge unique, since, as previously discussed, CL microscopy is hampered by the shadowing effect of the tip.

2. The manuscript has some grammatical/expressional errors, for example:

- line 18: shouldn't it say "... plasmonic modes to the photonic density of states..." ?
- line 37: "... has focused on the spectral characterization..."
- line 87: "...main peaks within an energy range from..." (without comma)
- line 99: "...cutoff at a/the maximum photon energy..."
- line 153: "...As a difference with.." is not a good expression for a comparison. It should say something like "Compared to...", or "in contrast to..."

All these minor typos/errors have been corrected in the new version of the manuscript.

3. The description in line 41 is not unambiguous: "...able to map the radiative...". Map spatially, spectrally? This needs more detailed discussion.

This sentence has been entirely removed in our current version of the introduction.

4. Some references need revision and more careful literature research, for example

- line 35: there has been a lot newer work published since 2013, with great achievements and high novelty. Also, the list in line 35 misses novel work on strong coupling in plasmonic cavities, which is directly relevant for the submitted work.
- line 49: Some relevant references to EELS studies are missing (e.g. <https://doi.org/10.1021/nn900922z>, or <https://doi.org/10.1021/nl5043775>)
- line 37: this sentence needs a reference (what techniques, what are the results?)

Our new introduction contains new citations (references 11, 14 and 17) to cover the different aspects that Referee 2 considered were missing in the previous version of the manuscript.

5. The authors claim that EELS is problematic for purely metallic samples because of large scattering of the electron beam. But in Ref. 13 and other work, metallic samples are investigated, partially also combined with CL (e.g. Losquin et al. 2015). Could the authors provide some evidence/reference for the stated problem of strong scattering, and what this implies?

The referee is right in pointing to this statement from our previous version of the manuscript as our sentence was too broad in scope. EELS can be and has been used in combination with CL to discriminate between bright and dark plasmonic modes in metallic nanostructures. What we actually intended to convey was that EELS cannot be used for these particular metallic nanocavities with a tip-on-surface geometry because the large scattering suffered by the electron beam precludes a normal incidence, and thus prevents the characterization of the plasmonic modes to which vertically oriented dipole light sources (quantum emitters) would couple. We thank Referee 2 for noticing this overstatement, which has now been rewritten in the new version of the introduction as: "the large scattering experienced by the probing electron beam when penetrating thick metallic regions, which precludes the incidence of the electron beam from the direction normal to the gap and, thereby, the excitation of gap modes, whose dipole moments are oriented perpendicular to the gap.." (lines 67-70).

6. A concept image/experimental scheme in Figure 1 would be nice and helpful.

Figure 1 now includes such a scheme as described by Referee 2. The caption and the main text (lines 81-84) have been modified to include its description.

7. How robust is the fitting of the peaks in the STML spectra, especially regarding the peak position of the 2.6 eV peak? The low-energy wing of the spectra below 2.5 eV, as for example that at 3.3 V in Fig. 1 c), seems very structured (with potentially more than two peaks underneath), and it is questionable if this can be fitted unambiguously by two gaussians. How sensitive is the factor 2.2 to the fitting? As the relevant shifts are in the few 10 meV range, robustness of the fit needs to be carefully discussed.

As the referee points out, our experimental spectra show noticeable structure at energies below 2.5 eV. In order to address the robustness of our fitting, we have tried to include one more peak in this energy region. The analysis is now shown in Figure S1, and discussed in the first section of the Supplementary Information. Including the extra peak indeed improves the quality of the fitting, but the effect on the main peak positions is small (completely negligible for the high-energy 3 eV peak, and of only 3-5 meV in the 2.5 peak). More importantly for the present work, the shifts with stabilization voltage remain unchanged both for the normalized and raw spectra. We thus conclude that the fitting procedure is quite robust towards reasonable choices of our fitting functions.

8. The authors model the tip-sample nanocavity by a sphere on a flat substrate. The STM tip, however, is not a closed nanoparticle but a half-terminated structure, which might support very different plasmonic modes. Moreover, STM tips often exhibit small nanoprotusions on a larger tip apex due to tip conditioning, which deviates strongly from the employed sphere model. How much are the results dependent on the geometrical shape of the nanoparticle? Would e.g. a nanorod or conical nano-cylinder yield similar results? How much do the gap modes depend on the semi-infinite tip shaft? The authors should justify the use of a sphere model and provide evidence that this yields comparable and correct plasmonic modes.

In order to address this sensitive issue of our model, we have performed further theoretical calculations, now included in the third section of the Supplementary Information, and presented through Figures S4 and S5. In brief, the exact shape of the nanoparticle does not seem to play an important role, as can be evidenced by comparing the modes supported by the nanocavities formed by ellipsoidal nanoparticles with different aspect ratios (Figure S4). Moving from finite nanoparticles to semi-infinite tip geometries, however, can lead to substantial changes depending on the apex geometry. This can be seen in Figure S5, where we have modelled the tip as a cone with a spherical protrusion. It can be observed that when the distance between the center of the sphere and the apex of the cone is small (simulating a rounded conical tip), the far-field spectra becomes dominated by leaky propagating modes (not confined to the cavity). However, for sufficiently large sphere-apex distances (i.e., when our tip can be thought of as having a well-defined protrusion), scattering at the kinked areas prevents the coupling of the localized plasmonic modes at the gap cavity to these leaky modes propagating along the cone surface. Thus, the far-field spectra becomes once again very similar to that of our spherical model (Figure S5a). This effect is even enhanced with increasing tip angles (Figure S5b). From these results, we can safely conclude that relatively broad tips with small but well-defined protrusions should behave largely as finite nanoparticles, thereby supporting our initial model. Of course, a broad tip can have more than one single

protrusion, overlapping with each other in different ways, which explains the large variability of the observed experimental spectra depending on tip conditions.

9. Related to 7: The simulations yield a proportionality factor of 1.15 between the spectral shift of the dipolar and quadrupolar modes. How does this factor depend on the geometry (e.g. sphere vs. nanorod)? Does this provide a valid (universal) reference for the tip geometry? As the main conclusion of the manuscript relies on the comparison of the experimentally observed shifts to that number, this needs to be clarified.

As described in our comments to the previous point, more realistic geometries consisting on a broad tip with a small protrusion display essentially the same spectra as geometries in which the whole tip is replaced only by the protrusion, so the peak shifts with distance should be the same.

10. (How) do the number of peaks and their spectral distribution vary for different tips? Given that the manuscript intends to provide a universal 'recipe' how to use STML to probe radiative plasmon modes in nanocavities, its sensitivity on the exact geometry and tip condition should be discussed, and reproducibility of the procedure for at least a second tip needs to be shown.

The variability of the experimental spectra with different tip conditions is rather large. As mentioned in the main text, most of the experimental spectra display one or two peaks in the range of energy between 1.5 and 3 eV, but some examples can be found with more than two peaks. Moreover, very often a finer structure can be observed in the shape of shoulders in the spectra that might reflect other plasmonic modes. The normalization procedure, however, removes quite efficiently the dependence of the spectra on the tunneling conditions in all studied cases. This is clearly shown in the new Figure S6 in the Supplementary Information. For this tip, we observe two relatively narrow peaks at 2.0 and 2.1 eV on top of a relatively broad intensity in the range of 1.6-2.2 eV. While the raw spectra show a rather marked dependence on the stabilization voltage, the normalization procedure makes the spectra indistinguishable.

11. The title suggests experimental probing of the radiative phDOS, but it is poorly discussed in the manuscript how the STML spectra are connected to the phDOS. The authors should explain more clear how STML accesses the (radiative) phDOS.

As discussed above, the radiative PhDOS can be characterized through the electromagnetic power radiated by the nanocavity into the far-field. According to Equation (5), it can be experimentally obtained for each photon energy by means of our normalization procedure. We have modified the discussion of the Equation (5) to further clarify this issue (lines 249-262).

12. The authors introduce plasmonic pseudomodes to explain the peaks in the simulated phDOS (starting line 153). Whereas the comparison of STML spectra to the simulated far-field spectra is obvious and clear, the connection to the phDOS and the role of the pseudomodes for SMTL remains vague. A clear connection of the pseudomodes and phDOS to the STML/far-field spectra and a detailed discussion of its relevance for the experimental data should be given.

As previously discussed, STML is not sensitive to the dark modes and, therefore, this technique is not sensitive to the plasmonic pseudomodes that govern the total PhDOS. Such plasmonic pseudomodes, however, control the Purcell enhancement and associated

fluorescence lifetime reduction, which can be accessed by optical means. By showing the total PhDOS in the insets of Figure 2a and 2b we aim at making clear that with STML we only observe the radiative modes, and such information is complementary to, but different from, the information that can be retrieved by a purely optical characterization.

13. An explanation of the gray errors in Fig. 2 should be given in the figure caption.

The gray arrows correspond to the energy positions for which the electric field intensity and charge density maps in Figure 2c are plotted. This is now described in the last sentence of the caption of Figure 2.

14. The phDOS (far-field spectra) in Fig. 2b) are scaled by δ^9 (δ^8). Is there a physical meaning of these exponents? Or is this used based on empirical findings? Such numbers should be explained properly.

There is not any physical meaning to those exponents. These factors are simply chosen to allow the direct comparison among different spectra.

15. The authors claim that the dependence of the inelastic transmission function on the photon energy can be neglected. However, one could expect that the probability to excite a plasmon via a tunneling electron depends on the available photonic density of states of the nanocavity, which is a function of the photon energy. In that sense, the authors need to justify this assumption, and it needs to be explained why this dependence is (significantly) weaker than the overlap of initial and final states.

Referee 2 is right when stating that the probability for inelastic tunneling should be larger for energy losses corresponding to plasmon energies with a high PhDOS, but this effect simply adds the PhDOS as a new factor to the rate without changing the transmission. This can be simply obtained from Fermi's Golden Rule as follows: we consider our base states to include the occupation numbers for the electron states at the tip, the electron states at the sample and the plasmon states. An inelastic tunneling event is one in which one electron is annihilated, say, from an electronic state at the tip with a given energy, created at an electronic state at the surface with a lower energy and a plasmon with energy equal to the difference in electronic energies is created. According to Fermi's Golden Rule, the rate of such inelastic events for a given plasmon energy is simply proportional to the sum of the squared matrix elements of the Hamiltonian to all such initial and final many-body states. By replacing the sum over states to an integral over energies, one obtains Equation (5), allowing for the identification of the transmission factors with the squared matrix elements of the Hamiltonian between initial and final states. Because these matrix elements are referred to well-defined many body states, they can be calculated without consideration of how many states are there at one particular energy and, therefore, they are independent of the electronic and plasmonic DOS, as we stated above. We have included this discussion in a somewhat more pedagogical way in our new version of the manuscript. The rate of inelastic transitions is now defined for a hypothetical case of a plasmonic mode with perfectly defined energy, i.e., the photonic density of states is a delta function (lines 202-213). The rest of the argument remains the same, and we arrive to the conclusion that the rate of inelastic transitions in this hypothetical case should just be the tunnel current at a different voltage. Notice that for this situation there can be no dependence of the transmission functions on the PhDOS. Now, for the more realistic

situation in which the PhDOS is a relatively broad and structured function, we just apply Fermi's Golden Rule and conclude that the real inelastic current would just be the product of the previously obtained rate of inelastic transitions times the PhDOS. The light intensity thus results from considering that not all the plasmonic states radiate with equal efficiency, and thus we should substitute the total PhDOS for the radiative power of the cavity that characterizes the radiative density of optical states (lines 249-262).

16. At the end of the results section it has been explained that the STML spectra after normalization are governed solely by the radiative plasmonic modes in the nanocavity. Again, it however remains unclear to the reader how this connects to and unveils the radiative phDOS, as suggested in the title. In that regard the title is not chosen adequate enough.

As it has been discussed in previous points, the radiative PhDOS is characterized by the radiative power of the nanocavity and, according to Equation (5), this is precisely what is obtained from our normalization process.

17. The discussion part would benefit from a more detailed discussion on the wide-ranging implications of the results for the understanding of the photonic and plasmonic properties of nanocavities (also compared to other methods), beyond the technical achievement to use STML as a probe for the radiative properties of such nanocavities.

Our findings do not only show that STML is a valid characterization tool for plasmonic nanostructures. In addition, they reveal that STML provides experimental insights into the key, and up to our knowledge unexplored, question of the role of radiative optical modes in archetypal gap nanocavity geometries. We believe that this point is clearly made in the new version of the manuscript.

Reviewer #3 (Remarks to the Author):

In this manuscript, the authors report an experimental procedure to evaluate optical response of a plasmonic nanocavity formed between an STM tip and metal substrate. They showed bias voltage dependency of STML spectrum and explained the main spectral features with the aid of theoretical EM simulations. Then they discussed the influence of electronic structures of the STM tip and metal substrate on STML spectra to derive a simple expression for rate of inelastic tunneling (4). Finally, they demonstrated that STML spectra can be very easily normalized to show only optical properties of the plasmonic nanocavity which is independent of the electronic structures of the STM junction.

The major novelty of the present manuscript lies in establishment of an easy method to disentangle the optical and electronic properties of a nanocavity, which can be utilized widely in this kind of experiment. The demonstration is remarkable to my perspective since the extremely small size of the electromagnetic field at the nanocavity usually makes it very difficult to understand the nature of the field itself, and, at the same time, the small EM field is the source of many intriguing application of plasmonics. For deeper understanding of nanocavity plasmons STM combined with an optical system is a promising platform. With the achievement in this manuscript I expect we start to understand the nature of the important EM field.

I recommend publication of this article in Nature Communications after my criticism listed below has been taken into account.

We thank Referee 3 for his/her positive response to our manuscript. We do believe that the results presented in this paper can become a cornerstone for the understanding and analysis of STM-based luminescence experiments.

1. I believe the most important part of this study is the normalization procedure, so I recommend to put Figures S1 and S2 into the main text.

Following referee's recommendation, a new figure (Figure 4) has been added to the main text of the manuscript. This new figure includes the information provided by Figures S1 and S2 of the previous version of the Supplementary Information. In accordance, part of the discussion of those figures has also been moved from the Supplementary Information to the main body of the paper (lines 243-248).

2. To generalize the findings, several different data sets (different tip conditions) should be added. This also contributes to making the effectiveness of the method clearer.

3. In line with the previous comment, raw experimental data and normalized data measured at both negative and positive voltages should be added and discussed.

We have added a new section to the Supplementary Information including the luminescence spectra from a different tip at negative voltages, and the result of the normalization process. It can be clearly seen in Figure S6 that the normalization yields excellent results regardless of the specific tip conditions and bias polarity.

4. The authors suggested 5 nm tip radius based on the EM calculations. However, 5 nm sound too small for a radius of an STM tip and I believe the real STM tip radius is much larger than 5 nm. If it is possible, addition of an SEM image showing the tip radius is desirable, or adds some comments on discrepancy between the experiment and theoretical simulation.

Referee 3 is completely right, and our own SEM characterization of the tips yield radii of the order to several tens to hundreds of nanometers. It is however well known that STM tips also support nanometric or even atomic-scale protrusions that actually play an essential role for the spatial resolution achieved routinely with STM. Based on our new calculations, we now believe that these protrusions can largely determine the optical properties of the nanocavity. This is clearly shown in Figure S5: here we have modelled our tip as a cone with a spherical protrusion, and characterized the far-field light intensity as a function of the geometrical parameters of the system, i.e., the distance from the center of the sphere to the apex of the cone and the opening angle of the cone. Our results show that, when this distance is small and the geometry corresponds mostly to a "rounded" cone, excitation of the electromagnetic field at the nanocavity leads to propagating surface plasmons along the surface of the cone. These modes are leaky and, thus, dominate the far-field spectrum, which is now very different from that of the sphere alone. On the other hand, when the distance is large, a geometry that corresponds to a nanometric protrusion on the apex of the tip, scattering at the kinked area between the cone and the sphere precludes the coupling between the localized plasmonic modes at the cavity and the propagating modes at the surface of the cone, and the spectra becomes very similar to that obtained for an isolated nanoparticle with nanometer-scale radius. The fact that the plasmonic modes are dominated by the atomic- or nanometer-scale protrusions is also probably related to the large variability of the experimental far-field spectra recorded by

us and other groups, since such protrusions are not very stable and can be modified by tip reshaping procedures while scanning.

5. It is a well-known fact that Ag(111) surface has a surface electronic state located around 50 meV below the Fermi level, and it can be expected that the surface state play a role in the inelastic tunneling process based on the conclusions of this work. I believe that the quality of this work would be considerably improved if the authors can show the signature of the surface state in STML spectrum.

Following referee's comment, the role of the surface state in Ag(111) has also been addressed in Figure S6. At negative voltages, the surface state can be clearly identified in the dI/dV spectra as a step in the conductivity at about -45 meV. This stepwise evolution of the conductivity is related to a well-defined kink in the $I(V)$ curve. Since we obtain the rate for inelastic transitions directly from the $I(V)$ curve, we expect it to have a kink at the photon energy $|V_{\text{bias}}+45\text{meV}|$ (remember V_{bias} is negative now). This kink is difficult to see clearly though, because it needs to be multiplied by the radiative power of the nanocavity which also shows a non-trivial dependence on the photon energy. However, the data from Figure S6 seem to be good for the observation of the effect of the surface state, since both sharp peaks at 2 and 2.1 eV have the very similar intensities upon normalization. Thus, by choosing a bias voltage slightly above the position of the higher energy peak, we can expect to see the kink in the intensities of the raw spectra, and this is indeed the situation.

Minor comments;

1. The first sentence in the abstract is not easy to understand, especially to general people. Probably because too much jargons are used.

In accordance with referee's suggestion, we have changed the first part of the abstract in order to be understood by a general reader.

2. I recommend to include a schematic diagram to Fig.1 which illustrate the experiment in a simple way.

Following referee's suggestion, Figure 1 now includes such a scheme as described by the Referee. The caption and the main text (lines 81-84) have been modified to include its description.

3. The optical system consists of three plano-convex lenses to lead the emitted light from a point underneath the STM tip to the detector. This is not a normal setup, because even number of plano-convex lenses should be used. It is helpful to includes a schematic diagram of the experimental setup.

4. I don't understand why two plano-convex lenses outside of the chamber have very long focal lengths, 300 and 200 mm.

The referee is right if we assume ideal lenses with ideal point emitters, but unfortunately, in our set-up we have geometrical constrictions and aberrations. The lens mounted on the STM head (lens 1) is far from ideal, due to its small diameter; and is subject of chromatic, spherical and coma aberrations. These constrictions along with the fact that our system does not allow for a mechanical repositioning of lens 1 (which sits inside the vacuum

chamber) to correct the focal point depending on tip position, preclude the formation of a perfect collimated beam out of the STM tip. We therefore opted to set the STM tip (point emitter) at a slightly longer distance than the focal length of lens 1, in such a way that an image of the point emitter is created inside the UHV chamber, very close to the optical window of the chamber. This allows us to create a collimated beam using lens 2, which is outside the UHV chamber and we can easily reposition to make their focal point match the position of the tip image formed by lens 1. In this way it is always possible to create a collimated beam. The last lens is employed to project the collimated beam into the entrance slit of the spectrometer. The focal lengths are determined by the geometric constrictions of our STM and vacuum chamber.

5. In the expression (S2), z should be δ .

This has been corrected in the new version of the manuscript.

6. In figure S3 (b) the vertical axis should be angstrom.

Referee 3 is completely right. Just to be consistent with our choice of units, we have changed the numbers in the scale to nanometers (nm).

Reviewers' Comments:

Reviewer #1:

Remarks to the Author:

The revised manuscript addresses the concerns made by all reviewers. In my view the manuscript is now in a form suitable for publication in Nature Communications.

Reviewer #2:

Remarks to the Author:

Reply to the revised manuscript "Unveiling the Radiative Local Density of Optical States of a Plasmonic Nanocavity by STM Luminescence and Spectroscopy" by R. Otero and coworkers.

I thank the authors for their detailed and instructive response to the points raised. I am happy to write that all issues are clarified and I am very much satisfied with the quality of the revised manuscript. The re-written introduction is now very nice to read and follow and nicely highlights the importance of this work for the characterization of tip-based optical nanocavities. The manuscript introduces and convincingly verifies the validity of a purely experimental and simple procedure to eliminate the electronic contribution to STML spectra and thus to reliably characterize the radiative modes in a plasmonic STM junction via STML. I thank the authors for providing additional calculations investigating the influence of the tip geometry as well as applying the normalization procedure to an additional data set from a second tip, strongly supporting the robustness of the normalization procedure. Moreover, I appreciate very much the additional figure 4 in the main manuscript, supporting the discussion of the main equations 4 and 5 and strengthening the main aspect of the manuscript. I only have a few very minor comments:

- Fig. 4a): it would help to add to the caption that the dashed lines are an example for one specific photon energy.
- Fig. 4c): it should say "...with the same tunneling parameters as in b...". Moreover, it is not written what the dashed lines are (rate).
- Fig. 4b) and c): The meaning of the three circles is not mentioned anywhere, this should be briefly explained either in the caption or main text.

This work constitutes a great improvement in the understanding of STML spectra recorded from plasmonic nanocavities and opens up new possibilities towards the characterization of quantum emitters and molecules coupled to plasmonic cavities via STML. I recommend publication of the revised manuscript in Nature Communications.

Reviewer #3:

Remarks to the Author:

The authors have responded convincingly to my criticisms. Consequently, I recommend to publish now this nice manuscript.

**Subject: Reply to referees –
“Unveiling the Radiative Local Density of Optical States of a Plasmonic
Nanocavity by STM Luminescence and Spectroscopy”**

Dear Referees,

Once again we wish to express our gratitude for the constructive interaction that we had during the evaluation of our manuscript. We are glad to know that you all agree with the publication of our manuscript. In particular, we would like to thank Referee 2 for pointing out some final gaps in the caption of our Figure 4, which have now been filled.

Best regards

Roberto Otero & Francisco J. García-Vidal (on behalf of all the authors)